# A Review of the Enhancement of Bio-Hydrogen Generation by Chemicals Addition

**Yong Sun** [1,2,*] **, Jun He** [1] **, Gang Yang** [3,*] **, Guangzhi Sun** [2] **and Valérie Sage** [4,*]

1   Department of Chemical and Environmental Engineering, University of Nottingham Ningbo, Ningbo 315100, China; Jun.He@nottingham.edu.cn
2   School of Engineering, Edith Cowan University, 270 Joondalup Drive, Joondalup, WA 6027, Australia; g.sun@ecu.edu.au
3   State key Laboratory of Biochemical Engineering, Institute of Process Engineering, Chinese Academy of Sciences, Beijing 100190, China
4   The Commonwealth Scientific and Industrial Research Organization (CSIRO), Energy Business Unit, Canberra, WA 6151, Australia
*   Correspondence: yong.sun@nottingham.edu.cn or y.sun@ecu.edu.au (Y.S.); yanggang@home.ipe.ac.cn (G.Y.); valerie.sage@csiro.au (V.S.)

**Abstract:** Bio-hydrogen production (BHP) produced from renewable bio-resources is an attractive route for green energy production, due to its compelling advantages of relative high efficiency, cost-effectiveness, and lower ecological impact. This study reviewed different BHP pathways, and the most important enzymes involved in these pathways, to identify technological gaps and effective approaches for process intensification in industrial applications. Among the various approaches reviewed in this study, a particular focus was set on the latest methods of chemicals/metal addition for improving hydrogen generation during dark fermentation (DF) processes; the up-to-date findings of different chemicals/metal addition methods have been quantitatively evaluated and thoroughly compared in this paper. A new efficiency evaluation criterion is also proposed, allowing different BHP processes to be compared with greater simplicity and validity.

**Keywords:** hydrogenase; bio-hydrogen; chemicals addition; review

## 1. Introduction

To effectively curb the world emissions from fossil-based energy by 2030 [1,2], attempts of exploring alternative renewable energy have been made worldwide in both scientific and industrial communities in the past decades [3,4]. Due to its great features, such as having the highest energy density among other fuels and complete cleanness after combustion, hydrogen has attracted a lot of attention as an energy carrier [5]. However, the current existing hydrogen generation processes have been dominated by the conventional routes of natural gas steam reforming (SR), natural gas thermal cracking, coal gasification, and partial oxidation of the heavier-than-naphtha hydrocarbons, which use fossil fuel as feedstock, are energy-intensive, and less environmentally friendly [6,7]. Although direct water-splitting via a semi-conductive photocatalyst to produce renewable hydrogen has recently attracted much interest [8], the significant bottom neck of very low efficiency still remains a big technical hurdle to be overcome for its short- and middle-term industrial application. On the other hand, biological processes for hydrogen generation possess many intrinsic appealing advantages, such as simplicity in operation, wide availability of renewable feed stocks (such as agricultural waste and food waste), carbon neutrality, and cost-effectiveness in operation [9–14]. Bio-hydrogen production can be achieved by two kinds of biological processes: (1) light-dependent, and (2) light-independent. For photo-dependent

processes, it could be further divided into the photolysis and photo-fermentation subcategories. For the photo-independent processes, hydrogen generation is achieved by dark fermentation (DF). For all of these processes, the bio-catalyst hydrogenase ([FeFe], [NiFe], [Fe]) might be the most significant catalyst for the evolution of hydrogen. Among all of these above-mentioned processes, DF is one of the most promising due to its appealing features of simplicity of operation, relatively high hydrogen conversion, flexibility in cultivation, and simultaneous realization of hydrogen production and organic waste consumptions [15]. Therefore, the effective enhancement of BHP during DF has become a research focus among scholars in the last decades. Many approaches have been found to effectively enhance hydrogen generation during the DF, which include pretreatment (e.g., ultrasonic, acid/base, enzyme hydrolysis), optimized operation (e.g., hydraulic retention time), co-fermentation, genetic engineering, and chemical addition [16–18]. Some of those approaches will directly or indirectly affect the hydrogenase biocatalyst, while some others might affect the other metabolic pathways or the growth of microbes, which ultimately accelerate or inhibit hydrogen generation [19,20]. From a practical perspective, the chemical addition is more feasible compared with other approaches mentioned above. This is the reason why the numbers of reports in regard to process intensification by chemical addition have been growing very rapidly in recent years [21]. Therefore, this motivated us to review the recent progress of chemical addition, such as metal monomers, metal oxides, nanoparticles (NPs), and synergistic factors that potentially affected the activities of a hydrogenase biocatalyst and consequently led to increased hydrogen generation. The hydrogen production rates will be quantitatively compared among these different works. In this review paper, to avoid repetitive summary and discussion that had been addressed by other scholars, we only focus on chemical addition that could potentially affect the activity of hydrogenase during DF for BHP.

## 2. Enzyme System in Bio-Hydrogen Generation

Hydrogen generation via biological processes can be achieved by a series of biological electrochemical reactions. These reactions are facilitated by a series of biocatalyst enzymes that are found to play critical roles during the BHP. There are three main bio-hydrogen production and consumption enzymes, which are responsible for the net bio-hydrogen evolution. These three different enzymes are reversible hydrogenase, membrane-bounded uptake hydrogenase, and nitrogenase enzymes. Among them, nitrogenase and hydrogenase are the two pivotal biocatalysts [22].

### 2.1. Functions of Nitrogenase

Hydrogen generation can be catalyzed by nitrogenase under an anaerobic environment at photofermentation conditions from photosynthetic bacteria. Nitrogenase is well-known for fixing the nitrogen molecule, and is commonly found in archaea and bacteria. While the nitrogen molecule is catalyzed into ammonia by the nitrogenase, hydrogen gas is generated as a by-product, and the entire chemical redox balance is maintained during this biological catalytic nitrogen fixation process, which is summarized in Equation (1) below:

$$\text{N}_2 + 8H^+ + 8e^- \xrightarrow{\textit{Nitrogenase}} 2NH_3 + H_2 \uparrow \qquad (1)$$

Hydrogen generation catalyzed by nitrogenase is thermodynamically regarded as an energy-intensive and irreversible reaction, which consumes four moles of adenosine triphosphate (ATP) per mole of bio-hydrogen produced. Ammonia (product) removal and an anaerobic condition is critical to hydrogen generation. A schematic diagram of the structure of nitrogenase is shown in Figure 1.

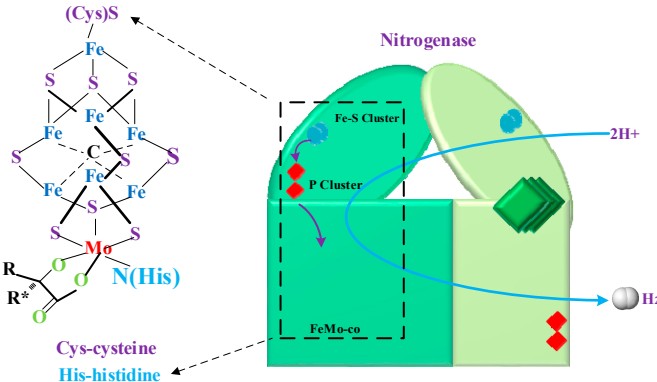

**Figure 1.** Schematic structure of nitrogenase, where R and R* are the ligands. The figure was rearranged from Seefeldt et al. [23].

The typical structure of nitrogenase consists of three metal-containing cofactors, which are the iron-sulfur cluster, P cluster, and FeMo cluster. The iron–sulfur cluster serves a critical role in delivering electrons to the FeMo cluster via the P cluster. The FeMo serves as an active site for dinitrogen reduction to ammonia. The nitrogenase enzyme widely exists in the photofermentation in archaea and bacteria. Factors such as chemical additions that either enhance or suppress the activity of nitrogenase will result in a variation of hydrogen evolution. Taking the purple non-sulfur bacteria (PNSB), for example, under nitrogen-deficient conditions, the turnover from the nitrogenase is continuous, reducing the protons into $H_2$. During each circle, at the Fe–S cluster associated with FeMo-co, 2 ATP are hydrolyzed with the transfer of one electron to the MoFe protein and the complex dissociates. The entire turnover is extremely slow at 6.4 s$^{-1}$, and added to its additional great deal of enzymatic machinery, energy, and time used for the biosynthesis of these complex metal centers, it consequently results in low efficiency [24].

## 2.2. Functions and Classification of Hydrogenase

The uncovering of the molecular structure of hydrogenase began from the first report of the atomic structure of the D.gigas enzyme [25]. The metal centers, which are the active sites of the biocatalyst, can be broadly classified into three different types, namely the [NiFe], [FeFe], and [Fe] types [26,27]. The [FeFe] hydrogenase catalyzes the oxidation of $H_2$, as well as the reduction of $H^+$, but the enzyme is mainly found in the $H_2$ generating process, whereas the [NiFe] hydrogenase catalyzes the consumption of hydrogen [28]. The detailed schematic diagram of the molecular structure of hydrogenase is shown in Figure 2. The active site of [NiFe] hydrogenase is a dinuclear thiolate-bridged Ni-Fe complex. The [FeFe]-hydrogenases' active sites are organized into modular domains with accessory clusters functioning as inter- and intra-molecular electron-transfer centers electronically linked to the catalytic H-cluster. Hydrogenases, especially the [FeFe] hydrogenase, are sensitive to the presence of oxygen (which is only active under strictly anaerobic conditions). However, studies have shown that the [NiFe] hydrogenases present better $O_2$ tolerance than the hydrogenase with [FeFe] metal centers [29]. In the metal center of a [NiFe] type hydrogenase, the active site usually contains two cis nickel (Ni) coordination sites available for substrate binding, a bridging site and a terminal Ni site, with the Ni site terminally bound to the thiolate of Cysteine 530 [30]. [NiFe] widely exists in bacteria during hydrogen fermentation, while the [FeFe]-type hydrogenase can only be found in a few microbial species, such as green algea Chlamydomonas reinhardtii [31,32].

Although the two biocatalysts—namely, the nitrogenase and the hydrogenase—show completely different structures and catalyze hydrogen generation via completely different reaction pathways, these two types of biocatalyst sometimes coexist within the cell of one microbe. Therefore, the addition of metal elements such as nickel or iron will affect the activity of these metal-based biocatalysts, which, in turn, will enhance or inhibit the hydrogen generation.

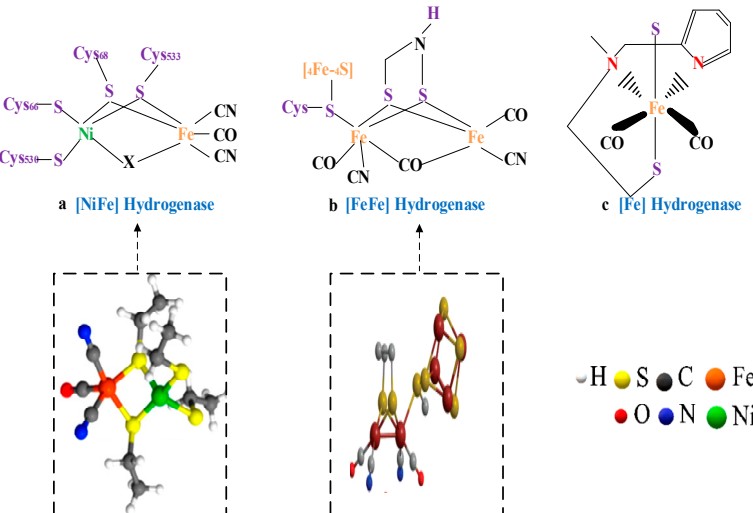

**Figure 2.** Schematic structure of different types of hydrogenase: (**a**) [NiFe] hydrogenase; (**b**) [FeFe] hydrogenase; and (**c**) [Fe] hydrogenase. The figure was adapted and rearranged from Fontecilla-Camps et al. [29].

## 3. Bio-Hydrogen Production Pathways

Bio-hydrogen production can be achieved by two major processes: (1) light-dependent processes, and (2) light-independent processes. The light-dependent processes can be accomplished by biophotolysis and photofermentation processes, while the light-independent processes can be realized by dark anaerobic fermentation. A conceptual illustration of the bio-hydrogen production pathway is shown in Figure 3.

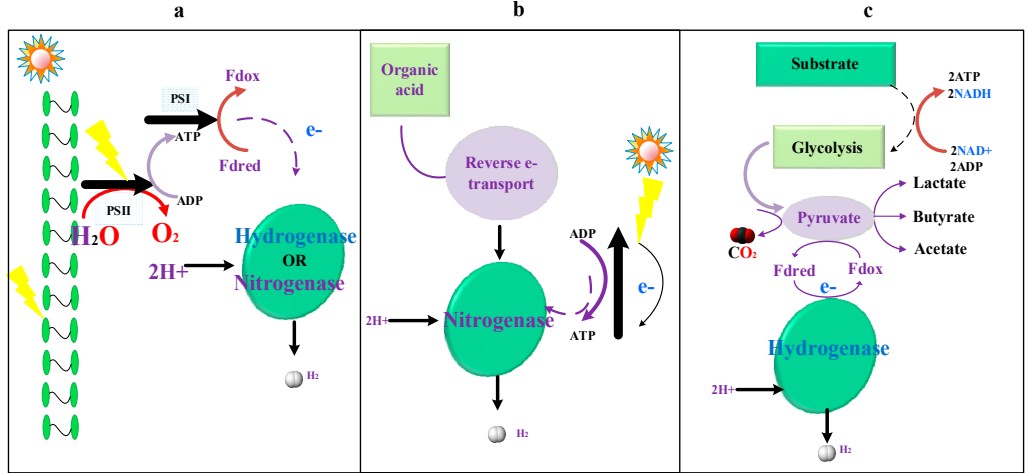

**Figure 3.** Conceptual illustration of the bio-hydrogen generation pathways: (**a**) biophotolysis, (**b**) photofermentation. (**c**) DF (dark fermentation), PSI represents photosynthesis system 1, PSII is photosynthesis system 2, Fdox is the oxidized Ferredoxin, and Fdred is the reduced Ferredoxin.

### 3.1. Biophotolysis Process

In biophotolysis (BP), the hydrogen ion is catalyzed either by nitrogenase or hydrogenase ([FeFe] [Fe]) to produce hydrogen gas in the presence of light within the cells of microbes (Figure 3a). Species such as algae and cyanobacteria ([NiFe]-type hydrogenase) are able to adopt this pathway to produce this zero-emission hydrogen gas from sunlight radiations [33]. BP can be further classified into two subcategories—direct biophotolysis, and indirect photolysis [34]. Many microbes, such as green algae or cyanobacterium, which are able to harvest solar energy to drive the water-splitting process to produce $O_2$ and reduce the ferredoxin-an electron carrier in the chloroplasts, are able to

perform biophotolysis via this direct BP pathway [35]. The water-splitting reaction is catalyzed by the photosynthesis system 2 (PSII) under anaerobic conditions, leading to the formation of hydrogen. The amount of electrons is linearly transferred from water to the ferredoxin, driven by the light energy harvested by PSI and PSII in the absence of oxygen. The reduced ferredoxin then donates the obtained electrons from PSI to the enzyme (hydrogenase or nitrogenase) to form hydrogen gas from protons. This entire pathway is shown in Figure 3a.

The microbes that are able to produce hydrogen via the BP process includes the following: *Chlamydomonas reinhardtii*, *Chlorella fusca*, *Scenedesmus obliquus*, *Chlorococcum littorale*, and *Platymonas subcordiformis* [36]. Due to the powerful suppressive effect of the oxygen as a by-product of PSII, the entire hydrogen generation process, including gene expression, mRNA stability, and enzymatic catalysis, will be strongly negatively influenced. Therefore, effective approaches in enhancing direct BP should focus on how to effectively remove or purge the oxygen produced from the system [37,38].

Another route of BP is indirect biophotolysis, of which oxygenic photosynthesis and hypoxic nitrogen fixation reactions are spatially separated from each other. Indirect biophotolysis is widely adopted by cyanobacteria. These are mostly filamentous, and nitrogen fixing in the specialized cell is known as heterocysts. Genera, such as Nostoc, Anabaena, Calothrix, Oscillatoria, are able to produce hydrogen via this indirect photolysis. Some non-nitrogen fixing genera, such as Synechocystis, Synechococcus, and Gloebacter, are also reported to possess this indirect BP pathway [33,36,39].

## 3.2. Photofermentation Process

In photofermentation (PF), the reduction of ferredoxins and generation of ATP is achieved via the reverse electron flow, driven by harvested solar energy, with the purple non-sulfur photosynthetic bacterium (PNS) under anaerobic conditions [40,41]. Instead of obtaining electrons from water-splitting reactions, as in the direct photolysis that exists in microalgae or cyanobacteria, the organic compounds, such as organic acid, acts as an electron donor under anaerobic conditions in the PNS bacterium. The schematic diagram illustrating this indirect photofermentation is shown in Figure 3b. The hydrogen generated via this pathway has appealing advantages: (1) complete substrate conversion to $H_2$ and $CO_2$; (2) removal of the adverse effect of oxygen that inhibits the activity of [FeFe] hydrogenase, hoxEFUYH [NiFe] hydrogenase, and nitrogenase enzymes [42–45]; (3) effective utilization of sunlight in both visible (400–700 nm) and near-infrared (700–950 nm) regions of the solar light spectrum; (4) wide availability of organic compounds used as an electron donor; (5) a relatively lower energy barrier to overcome, compared with water-splitting in direct photolysis [46]. Species that are able to produce the hydrogen via this photofermentation process include the following: *Rhodospirillum rubrum*, *Rhodopseudomonas palustris*, *Rhodobacter sphaeroides*, and *Rhodobacter capsulatus* [47,48].

## 3.3. Dark Fermentation Process

The essence of dark fermentation (DF) is the catalytic reaction of converting organic substrates into hydrogen under anaerobic conditions. Instead of harvesting the energy from solar light, the energy used to drive the neutralization reaction between the proton ($H^+$) and electrons ($e^-$) to form hydrogen comes from the microbial metabolic oxidation of organic substrates, such as glycolysis of glucose to the pyruvate intermediate. Complicated metabolic products are produced during DF. The product distribution of metabolic products varies significantly with the varieties of microbes, the oxidation of the substrate, and the environmental conditions, such as pH, hydrogen partial pressure, and level of nutrition [49,50]. Taking the glycolysis pathways as an example (Figure 3c), the ATP is generated through the substrate phosphorylation and energy-yielding reactions, including the formation of redox equivalents, such as the reduction of oxidized nicotinamide adenine dinucleotide ($NAD^+$) to nicotinamide adenine dinucleotide (NADH). The produced pyruvate intermediate is then reduced by the produced redox equivalents to form intermediary metabolites, and eventually leads to lactate, $CO_2$, and ethanol formation [51]. Another fermentation pathway includes the transformation of pyruvate to acetyl-coenzyme A (Acetyl-CoA), accompanied by the formation of an additional redox equivalent,

$CO_2$ and formate, and eventually leading to the splitting of Acetyl-CoA, and generation of ATPs and acetate [52,53]. The routes for forming molecular hydrogen can be expressed by Equations (2)–(4):

$$C_6H_{12}O_6 + 2NAD^+ \rightarrow 2CH_3COCOOH + 2NADH + 2H^+ \tag{2}$$

$$2NADH + H^+ + 2Fd^{2+} \rightarrow 2Fd^+ + NAD^+ + 2H^+ \tag{3}$$

$$2Fd^+ + 2H^+ \rightarrow 2Fd^{2+} + 2H_2 \tag{4}$$

where Fd represents ferredoxins.

These hydrogen-generation reactions are catalyzed by hydrogenase under an anaerobic condition. There are many appealing advantages of generating hydrogen via these DF pathways: (1) relative simplicity in hydrogen generation, with higher conversion, production efficiency, and lower energy input; (2) versatile feedstock, such as organic food waste or inorganic waste, used for the fermentation; (3) the anaerobic conditions will create a favorable state for maintaining better activity of the biocatalyst for both [NiFe] and [FeFe] hydrogenase, and result in a relatively larger yield of hydrogen; (4) the bio-hydrogen fermentation process is flexible to create either a pure or mixed cultivation of the microbes. A brief comparison of different bio-hydrogen pathways, their corresponding technical challenges, and their effective approaches for hydrogen generation enhancement are summarized in Table 1.

**Table 1.** Summary of bio-hydrogen pathways catalyzed by hydrogenase.

| Pathways | Challenges | Microbes Strains | Hydrogen Enhancement |
|---|---|---|---|
| BP | - Low light conversion efficiency [54]<br>- Incompatibility to simultaneously produce hydrogen and oxygen [55]<br>- High cost for product removal (impermeable hydrogen bioreactor) [56] | - *Scenedesmus obliquus Chlamydomonas reinhardii* (green algae) [57]<br>- *Anabaena variabilis* (cyanobacteria) [58] | - Simultaneous separation or removal of aversive effect to the hydrogenase produced from oxygen [59,60]<br>- Co-culture optimization [61] |
| PF | - Low photo chemical efficiency [62]<br>- Relative lower hydrogen productivity [63]<br>- Higher energy demand required from nitrogenase | PNS bacteria, such as *Rhodopseudomonas genus* [64] | Process optimization, such as:<br>- Batch cycled arrangement [65]<br>- Recombined DNA techniques [66]<br>- Immobilization of microbes [67]<br>- Chemical additions, such as Ni, EDTA, DMSO [68] |
| DF | - Relatively poor yield [69]<br>- Metabolic products inhabitation [70]<br>- Lack of research using continuous fermentation | - Thermococcus onnurineus<br>- Enterobacter asburiae<br>- Bacillus coagulans<br>- Thermotoga neapolitana<br>- *Clostridium* sp<br>- *Escherichia coli*<br>- *Bacteroides splanchincus* [51,71,72] | - Hybrid cultivation [73,74]<br>- Chemical additions, such as metal monomers, metal ions and metal oxides [21,75]<br>- Nanoparticles [76]<br>- Membrane reactor [77] |

Although technical hurdles and challenges still remain for these three hydrogen generation routes, the DF is still among the most promising technical route for BHP, which attracts great research interests and has even been successfully established at a pilot scale [26,78]. Therefore, DF will be the focus of our subsequent discussion for the enhancement of BHP by chemical additions.

## 4. Metal Additives

Although many attempts at process intensification, such as pretreatment, process optimization, and co-fermentation have been found to be effective in enhancing hydrogen production, the supplementation of additives have attracted much attention in DF due to its simplicity and cost-effectiveness compared with other approaches of process intensification [21]. Among different kinds of supplementation of additives, metal additives are among one of the most widely employed. It has been found that trace metals play a significant role during the anaerobic fermentation process, especially for the activities of the hydrogenase [79]. The addition of metal into fermentation media has been identified to have the following beneficial effects: (1) facilitation of intracellular electron

transportation, and (2) provision of essential nutrition for microbial growth. In this paper, attention will be focused on the effects of metal addition on bio-hydrogen generation. For convenience of discussion, the chemical additions are further divided into subcategories, including monomer, metal ion, metal oxide, and others, such as chemical addition, together with the combination of different types of operations, such as immobilizations.

### 4.1. Metal Monomers

The addition of metal monomers, such as $Fe^0$ and $Ni^0$ during DF, were found to be able to enhance hydrogen generation. The effects of these added metal monomers could be broadly divided into the two categories: (1) directly affects the activity of the biocatalyst; (2) affects the complicated metabolic pathways during DF that leads to enhanced hydrogen generation. Results for the addition of various metal monomers are shown in Table 2. With the addition of different metal monomers, the hydrogen yield was enhanced by different factors, from 10% to 110%, compared with that of the control test without metal addition, depending on the specific conditions such as different inoculum, substrates, or fermentative conditions.

**Table 2.** Summary and comparisons of bio-hydrogen production with addition of metal monomers and nanoparticles.

| Metal | Conc/mg L$^{-1}$ | Feed | Organism | Process | Temp/°C | Yielda/ | Reference |
|---|---|---|---|---|---|---|---|
| Au (NPs) | 10 nM | Sucrose | MC | Batch | 35 | 4.47 (+61.7%) | [80] |
| Ag (NPs) | 20 nM | Glucose | MC | Batch | 37 | 2.48 (+67.6%) | [81] |
| Ni$^0$ | 2.5 | Glucose | AS | Batch | 37 | 57 [c] (+79.8%) | [82] |
| Ni (NPs) | 5.7 | Glucose | AS | Batch | 33 | 2.54 (+22.7%) | [76] |
| Ni (NPs) | 60 | Wastewater | AS | Batch | 55 | 24.7 [b] (+22%) | [32] |
| Fe$^0$ | 2000 | OWM | AS | Batch | 30 | 102 [b] (+46%) | [83] |
| Fe$^0$ | 400 | Sucrose | AS | Batch | 30 | 1.2 (+37%) | [84] |
| Fe$^0$ | 550 | Sludge | AS | CSTR | 37 | 650 [d] (+10%) | [85] |
| Fe$^0$ | 100 | DS | AS | Batch | 37 | 26 [c] (+16%) | [86] |
| Fe (NPs) | 400 | Grass | CB | Batch | 37 | 65 [c] (+44%) | [87] |
| Cu (NPs) | 2.5 | Hexose | CA | Batch | 30 | 1.7 (−3.5%) | [88] |
| Fe (NPs) | 200 | SJ | AS | Batch | 30 | 1.15 (+62%) | [89] |
| Ni + Fe (NPs) | 37.5 + 37.5 | Starch | AS | Batch | 37 | 250 [b] (110%) | [82] |
| Ni (NPs) | 35 | Glucose | CB | Batch | 35 | 212 [b] (+32%) | [70] |
| Ni (NPs) + BC | 35 | Glucose | CB | Batch | 35 | 238 [b] (+49%) | [70] |

OMW: organic market waste: DS: dewatered sludge; SJ: sugarcane juice. MC: mixed consortia; AS: Anaerobic sludge; CB: *Clostridium. Butyricum*; CA: *Clostridium acetobutylicum*; [a] mol/mol of hexose; [b] L/kg TSS or COD or VSS (TSS: total suspended solids, COD chemical oxygen demand, VSS volatile suspended solids); [c] mL/g-dry grass; [d] ml/L.d.

Among the different types of metal monomers, the iron metal monomers are the most promising due to their appealing advantages of relative low cost, and effectiveness in affecting the activity of hydrogenase [21]. In addition, the oxidative-reductive potential (ORP) of fermentation solution could be reduced by the addition of zero-valent iron, which in turn creates a thermodynamically favorable environment for the growth of bacteria. Besides zero-valent metal monomers, the addition of nanosize zero-valent metal monomers, such as iron, nickel, or gold nanoparticles (NPs) began to attract attention due to the unique surface size and quantum size effect. The addition of iron or nickel NPs will facilitate the acceleration of electron transfer between the ferredoxin and hydrogenase to drive hydrogen generation. In addition, the added zero-valent Fe or Ni NPs could be oxidized into metal ions, such as $Fe^{2+}$ or $Ni^{2+}$, via the anaerobic corrosion process, which will potentially produce very similar beneficial effects upon BHP as those metal ions of $Fe^{2+}$ or $Ni^{2+}$ addition do during the fermentation.

Based upon current reports, one of the highest improvements in hydrogen generation (+110%) could be achieved by adding Ni (37.5 mg/L) and Fe (37.5 mg/L) NPs together during the DF [82]. This surely indicates that the improved electron transfer enhances the overall activity of hydrogenase. However, instead of continuously enhancing the hydrogen generation, an overdose of the metal

monomer starts to produce adverse effects upon hydrogen production, as reported in previous research [70]. This indicates that too high a concentration of metal monomers can be harmful for both the activities of hydrogenase, and for other metabolic pathways that indirectly affect hydrogen generation. Therefore, the optimal condition that meets both maximum performance and cost-effectiveness of operation exists. From Table 2, it is not difficult to identify the existing challenges and limitations: (1) there is no consistent quantitative evaluation standard for the assessment of BHP, leading to difficulties in comparing the performance of the different metal monomers; (2) most of the DF focuses on using the sugars, such as glucose or sucrose, and there are very limited efforts at investigating the effect of an addition of metal monomers to the biocatalyst hydrogenase using other types of organic substrates, such as food wastes; (3) the operation of DF was mostly conducted in batch operation, which ends up with continuous inhibitory intermediates accumulation during the DF [90].

## 4.2. Metal Ions

Metal ion is one of the most common additives that could be used to enhance the catalysts' performances during the DF. The iron ion is widely employed, not only because of its relative cost-effectiveness compared with other metal ions, but also because of its essential role in the constitutions of hydrogenase and ferredoxin. Like the functionality of iron ions, the role of nickel ion in enhancing the activities of hydrogenase also appears to be obvious. According to the works reported by Grafe and Friedrich, the nickel ion has been found in several hydrogenase and nickel-dependent uptake hydrogenase [91]. According to Zhang et al. [92], the addition of nickel ion directly stimulated the activity of hydrogenase. According to the different hypothesis available, the availability of nickel to a cell may affect the activity of the biocatalyst itself or affect the synthesis of other protein, which, in the end, will contribute to the enhancement of hydrogen evolution [91,92].

A summary of the usage of different metal ions, mainly $Fe^{2+}/Fe^{3+}$ and $Ni^{2+}$, as additives during DF using different types of substrates, such as sugars, wastewater, and food waste is shown in Table 3. From this, it can be seen that the addition of metal irons, especially $Fe^{2+}$ and $Ni^{2+}$, are effective in enhancing bio-hydrogen fermentation. The role of the metal irons, such as $Fe^{2+}/Fe^{3+}$ and $Ni^{2+}$, is found to facilitate both the increase of biomass (cell growth) and hydrogen production during the DF. From the work reported by Hisham et al. [93], it was found that the addition of metal elements, such as $Ca^{2+}$ or $Mg^{2+}$ metal ions, led to a significant decrease in hydrogen generation (−30%, −70%), while the biomass experienced a steady increase up to 40%. According to [94], by adding ferrous chloride during DF, the hydrogen generation was enhanced by 650% (increased to 130 ml.g$^{-1}$). Although the improvement compared to the baseline was significant in that work, the absolute value of hydrogen produced (650 mL of cumulative hydrogen production) was marginal compared with other literature reports (which is often over 1000 mL cumulative hydrogen production within the similar duration of cultivation) [90]. Apart from the addition of singular ion, the hybrid mixtures, such as Fe-Ni or Ni-Mg-Al (hydrotalcite), were also found to be effective in enhancing the hydrogen production. Their addition was found to increase the hydrogen generation to about 70–80%.

From the above discussion, it can be suggested that the roles of the different types of metal ions during DF are completely different. The addition of $Fe^{2+}/Fe^{3+}$, $Ni^{2+}$ or the mixture Fe–Ni seems to directly affect the activity of hydrogenase, and therefore the bio-hydrogen generation process could be directly manipulated by adding these types of metal ions. However, in regard to other metal ions, such as $Ca^{2+}$, $Mg^{2+}$, $Cu^{2+}$, $Na^{+}$, it seems that these metal ions tend to affect the growth of cell mass or indirectly influence other relevant metabolic pathways during the DF, of which no obvious hydrogen production improvement was observed by these types of metal ion additions.

**Table 3.** Comparisons of bio-hydrogen production with addition of nickel and other metal nanoparticles.

| Metal Ion | Opt/mg L$^{-1}$ | Organism | Feed | Process | Temp/°C | Yield[a] | Reference |
|---|---|---|---|---|---|---|---|
| $FeCl_3$ | 60 uM | Cyanobacteria | BG | Batch | 30 | 0.06 [b] (+25%) | [95] |
| $FeSO_4$ | 300 | AS | Glucose | Batch | 35 | 302 (+56%) | [96] |
| $FeSO_4$ | 300 | CB | Glucose | Batch | 37 | 2.4 [c] (+30%) | [97] |
| $FeSO_4$ | 25 | CB | Glucose | Batch | 30 | 408 (+4.3%) | [93] |
| $FeSO_4$ | 63 | AS | PS | Batch | 30 | 226 (52%) | [98] |
| $FeSO_4$ | 100 | HTS | Glucose | Batch | 35 | 2.6 [c] (+13%) | [99] |
| $FeCl_2$ | 353 | AS | Sucrose | Batch | 37 | 132 (+650%) | [94] |
| $FeCl_2$ | 50 | AS | Glucose | Batch | 37 | 216 (+23.4%) | [100] |
| $FeCl_3$ | 213 | EA | Glucose | Batch | 30 | 1.7 [c] (+55%) | [101] |
| $Ni^{2+}$ | 0.6 | CD | Sucrose | Batch | 35 | 2.1 [c] (+107%) | [102] |
| $Ni^{2+}$ | 0.2 | MC | Glucose | Batch | 35 | 2.4 [c] (+75%) | [103] |
| $NiCl_2$ | 0.1 | AS | Glucose | Batch | 35 | 289 (+55%) | [103] |
| $NiCl_2$ | 16 | AS | SW | Batch | 34 | 1120 (+500%) | [104] |
| $MgCl_2$ | 200 | MC | Glucose | FB | 35 | 1.75 (+600%) | [105] |
| $Na_2CO_3$ | 2000 | AS | Sucrose | UASB | 37 | 40 (+300%) | [106] |
| $MgCl_2$ | 500 | CB | Glucose | Batch | 30 | 209 (−30%) | [93] |
| $CaCl_2$ | 500 | CB | Glucose | Batch | 30 | 82 (−72%) | [93] |
| NaCl | 5000 | CA | Glucose | Batch | 30 | 2.7 [c] (−29%) | [107] |
| Hydrotalcite | 250 | HTS | Sucrose | UASB | 37 | 3.4 [c] (+80%) | [108] |
| Ni-Fe | 50 + 25 | CB | Glucose | ACSTR | 30 | 300 (+70%) | [109] |

Hydrotalcite: Ni-Mg-Al; AS: anaerobic sludge: CB: Clostridium. Butyricum; HTS: heat treated sludge; EA: *Enterobacter aerogenes*; MC: mixed consortia; CA: Clostridium acetobutylicum; CD: Cow dung; BG: BG11$_0$ media; PS: potato starch; SW: synthetic wastewater. FB: fed-batch; UASB: upflow anaerobic blanket; ACSTR: anaerobic continuous stirred tank reactor. [a] ml; [b] μmol.mg$^{-1}$ h$^{-1}$; [c] mol.mol$^{-1}$; [d] imL L$^{-1}$ h$^{-1}$.

## 4.3. Metal Oxide

Metal oxides play a very similar role as to metal ions. In recent studies, it has been found that the reduced size of metal oxides (nanoparticle size) will be favorable to the electron transfer between ferredoxin and [NiFe] or [FeFe]-based hydrogenase, which in turn accelerates the catalytic reactions of hydrogen generation [110]. The summary of adding metal oxides and their corresponding BHP performance is shown in Table 4. Various kinds of metal oxides, such as $TiO_2$, CoO, $Fe_2O_3$, NiO, and their mixtures ($Fe_2O_3$/NiO), and substrates such as glucose, organic wastewater, glucose, and starch, were used in previous studies, conducted mostly in batch operation under mesophilic conditions. The addition of $NiO_2$ NPs, together with a co-addition of other NPs, were also found to enhance hydrogen generation due to facilitations of the electron transfer between ferredoxin and hydrogenase [111]. Therefore, the addition of metal oxides, especially with nanoparticle size, is another effective approach in directly enhancing the activities and performances of hydrogenase, which, in turn, will boost BHP.

**Table 4.** Impact of metal oxide upon the activity of nickel-contained hydrogenase for hydrogen production.

| Metal Ion | Opt/mg L$^{-1}$ | Organism | Feed | Process | Temp/°C | Yield[a] | Reference |
|---|---|---|---|---|---|---|---|
| $TiO_2$ | 100 | BA | Glucose | Batch | 30 | 160 (+46%) | [112] |
| CoO (NPs) | 1 | AS | POME | Batch | 37 | 0.5 [b] (+10%) | [113] |
| $TiO_2$ | 300 | RS | SM | Batch | 32 | 1900 (+54%) | [114] |
| $\gamma$-$Fe_2O_3$ | 25 | AS | SB | Batch | 30 | 0.9 [c] (+62%) | [115] |
| $\alpha$-$Fe_2O_3$ | 63 | MC | Inorganic salt | Batch | 30 | 3.6 [c] (33%) | [116] |
| $Fe_2O_3$ | 175 | CA | CL | Batch | 37 | 2.3 [c] (+18%) | [117] |
| NiO (NPs) | 10 | MC | CDW | Batch | 37 | 13 [b] (+33%) | [118] |
| NiO (NPs) | 1.5 | BA | POME | Batch | 37 | 25 [b] (+15%) | [113] |
| NiO (NPs) | 100 | AS | GS | Batch | 35 | 2.1 [c] (+107%) | [119] |
| $Fe_2O_3$/NiO | 50/10 | MC | CDW | Batch | 37 | 17 [b] (+45%) | [118] |
| $Fe_2O_3$/NiO (NPs) | 200/5 | AS | DW | Batch | 37 | 19 [b] (+25%) | [120] |

BA: Bacillus anthracis; AS: anaerobic sludge: RS: Rhodobacter sphaeroides; MC: mixed consortia; CA: Clostridium acetobutylicum. POME: Palm oil mill effluent; SM: Sistrom's medium; SB: Sugarcane bagasse; CL: curry leaf; CDW: complex dairy wastewater; GS: glucose and starch, DW: distillery water. [a] mL; [b] μmol mg$^{-1}$ h$^{-1}$; [c] mol mol$^{-1}$; [d] mL L$^{-1}$ h$^{-1}$.

*4.4. Others*

A summary of BHP with synergistic effects are compared in Table 5. Microbial immobilization is one of the most widely employed approaches used to prevent biomass wash-out when hydrogen evolution rate (HRE) is low during continuous operation. The appealing advantages of employing microbial or cell biocatalyst immobilization include: (a) tolerance toward the perturbation of environmental factors, such as temperature, pH, and accumulation of inhibitory intermediates; (b) higher bio-catalytic activity; and (c) higher process stability [121]. Table 6 summarizes the performances of different microbial immobilizer additions during DF [56]. Clearly, with the implementation of immobilization, the BHP is enhanced at different levels. Many researchers have found that immobilization supports, such as activated carbon (AC) or biochar (BC), tend to form a favorable thermodynamic chemical redox potential, which makes the hydrogen generation catalyzed by the hydrogenase run more effectively [70].

**Table 5.** Comparisons of bio-hydrogen production with synergistic effects of adding biomass immobilization and metals.

| Additives | Opt/mg L$^{-1}$ | Organism | Feed | Process | Temp/°C | Yield[a] | Reference |
|---|---|---|---|---|---|---|---|
| AC | 200 | AS | Glucose | Batch | 60 | 1.77 [c] (+106%) | [122] |
| BC | 10 | AS | Food waste | Batch | 35 | 1475 [d] (+41%) | [123] |
| Fe$^0$ + AC | 100 | HTS | CBR | Batch | 30 | 83 [b] (+48%) | [124] |
| Fe (NPs) + CAB | 1000 | AS | PW | Batch | 38 | 298 [b] (+400%) | [125] |
| Gel | 0.2% (*w/v*) | EA | Glucose | Batch | 30 | 1.77 [c] (+80%) | [126] |
| PF | 2500 | AS | Glucose | Continuous | 37 | 0.6 [c] (+21%) | [127] |
| Sponge | - | EA | Starch | Continuous | 40 | 3.03 [c] (+37%) | [128] |
| Foam + Fe | 0.7 (g) | EA | Molasses | Continuous | 37 | 3.5 [c] (+77%) | [129] |
| Fe$^2$ ++ BC | 300 | HTS | Glucose | Batch | 35 | 234 [b] (+48%) | [130] |
| Ni (NPs) + BC | 35 | CB | Glucose | Batch | 35 | 238 [b] (+49%) | [70] |

AC: activate carbon; BC: biochar; CBR: corn-bran residue, PF: polyurethane foam. AS: anaerobic sludge; HTS: heat treated sludge; EA: Enterobacter aerogenes; CB: Clostridium. Butyricum. CBR: corn-bran residue; PW: potato waste. [a] mL; [b] mL g$^{-1}$; [c] mol mol$^{-1}$; [d] mL L$^{-1}$ h$^{-1}$.

Apart from the immobilization approach, the conditions of operation are also found to be effective in enhancing BHP. For example, the hydrogen production yield was higher in continuous BHP, compared with the batch operation [129,131]. This is possibly due to the effective removal of inhibitory metabolic intermediates, which creates a favorable chemical environment for hydrogenase to catalyze the hydrogen formation reaction [132,133]. Nonetheless, although continuous operation could be more appealing compared to batch operation for large-scale production, cells washing-out is one of the critical problems that need to be carefully handled during continuous operation. Furthermore, it is also interesting to find that synergistic effects, such as the addition of immobilized support, together with nanoparticles such as nickel NPs, have a positive effect on BHP. This suggests that these additions directly affect the activities of biocatalyst hydrogenase, the electron transport, and the endurance to environmental perturbation, which in turn boosts hydrogen generation during the DF.

*4.5. Results Comparison*

In this work, we tried to summarize all the relevant reported works on BHP that we cited in order to find out some quantitative trends on the basis of substrate conversion efficiency ($Y_{H2/S}$) expressed in the mole of hydrogen produced per mole of substrate in mol mol$^{-1}$, hydrogen evolution rate (HER) expressed in mmol L$^{-1}$ h$^{-1}$, and specific hydrogen production rate (qH$_2$) expressed in mmol g$^{-1}$ h$^{-1}$. It has been addressed by many scholars that the C-molar-based mass balance is necessary when hydrogen yield and rate are expressed during DF [78]. The failure to present mass balance and kinetic data can lead to poor quality assurance and difficulties in quantitative comparison for dark fermentative BHP. It is necessary to set up presentation standards for hydrogen yield and rate, for the convenience of communication and cross-referencing throughout the scientific community.

Due to the limited number of literature and some inconsistences in the presentation of yields and rates due to the omission of mass balances during DF in some reported works, we only used data from literature reports with complete mass balance and rate expressions, and made a limited number of comparisons in regard to $Y_{H2/S}$ versus HER, and $Y_{H2/S}$ versus $qH_2$ during DF. The results are presented in Figure 4. In this work, for the convenience of comparisons, we divided the chemical additions into the five different categories, which are metal NPs, metal monomers, metal irons, metal oxide, and others (metals other than iron or nickel), respectively. In addition, among these five different categories for BHP, the different kinds of synergistic effects, such as operations (continuous stirred tank reactor CSTR) or microbial immobilization, were all considered and counted. Within the same category, the $Y_{H2/S}$, HER, and $qH_2$ were averaged, then based upon the calculated mean values ($Y_{H2/S}$, HER and $qH_2$), the deviations were calculated using the following:

$$\bar{s} = \sqrt{\frac{1}{N-1} \sum_{i=1}^{N} (x_i - \bar{x})^2} \qquad (5)$$

where $N$ represents the numbers of sample size, $\bar{x}$ is the sample mean, and $x_i$ is one sample value. The detailed calculations of those values of $Y_{H2/S}$, HER, and $qH_2$, together with their corresponding deviations, could be found from previous reports [51,90]. In regard to the variation of the deviations calculated from those reported, there are many different factors that could be attributed to the large deviations observed with the different metal additions for BHP: (a) different microbial strains, (b) different ways of operation, such as batch or CSTR, (c) different substrates, such as glucose and sucrose employed during DF.

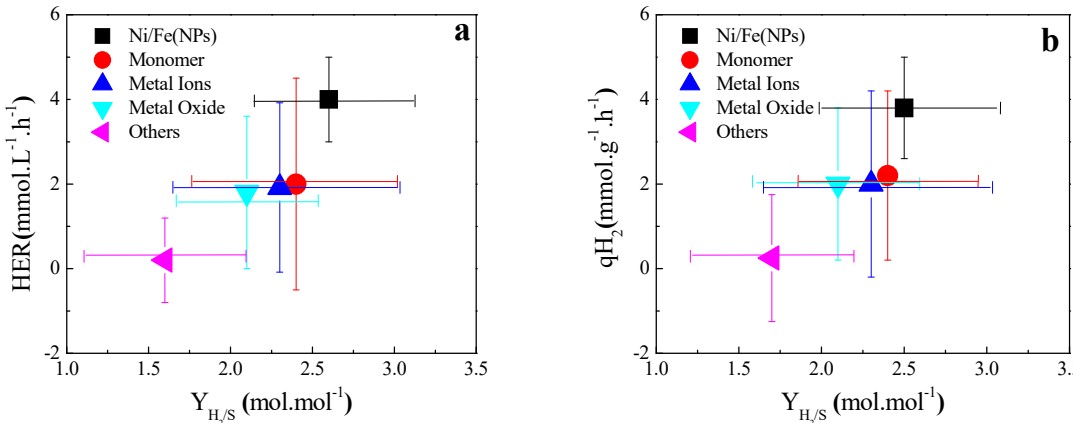

**Figure 4.** Data illustrating the mean and standard deviation of different metal additions, such as NPs, mental monomers, metal ions, metal oxides, and other metals on BHP during DF: (**a**) HER as a function of $Y_{H2/S}$; (**b**) $qH_2$ as a function of $Y_{H2/S}$. Note: Others exclude reports using immobilized supporters.

From the comparison, the enhancement of the activities of nickel contained hydrogenase can be broadly divided into three different regions. In the first region, the addition of nano-size particles (together with other synergistic factors, such as adding immobilized supports like AC or BC) is found to be relatively more effective in facilitating the bio-catalytic hydrogen reactions, on the basis of both substrate conversion efficiency and hydrogen evolution rate. The added NPs (such as Ni, Fe, or Ni/Fe) not only facilitate the electrons transfer, but also are engaged in other synergistic factors, such as immobilization and improved operations, that ultimately improve the BHP under anaerobic conditions [70]. In the second region, the metal monomer, metal ions, and metal oxides show very similar performances in enhancing the activities of nickel- or iron-containing biocatalysts during BHP. However, their effects tend to be complex: not only do they possibly affect the activities of the hydrogenase, but these additions might also affect other metabolisms or pathways and cell growth,

which will eventually contribute to the increase of BHP. The third region is for those metals other than Ni, Fe, which are relatively less effective in enhancing the BHP, and it was found that some of them are even toxic to either the activities of the hydrogenase or the growth of microbes. Therefore, the addition of these materials is not recommended for the enhancement of the activities of hydrogenase during BHP.

## 5. Economic Perspective of Different Hydrogen Generation Routes

At the present time, hydrogen is predominantly produced by thermal technologies on the commercial scale, via SR, partial oxidation (POX), and autothermal reforming (ATR). The most widely used and economical approach of hydrogen production is via the steam reforming of methane (natural gas) (SMR), which nearly accounts for 90% of the world's hydrogen generation, at a cost of U.S.$ 7/GJ [7,134]. One of the thorny challenges for these thermochemical processes lies in the simultaneous generation of greenhouse gases that needed to be captured and stored [135,136] or indirectly converting the produced $CO_2$ back into hydrocarbons via catalytic processes, such as Fischer-Tropsch (FT) synthesis [137,138], which will inevitably increase the cost of the entire hydrogen generation process by about 20 to 40% [139]. The alternative route of replacing fossil-based hydrocarbons with carbon-neutral biomass leads to a doubling of the cost of hydrogen production (about US $14–15/GJ, depending on the types of feedstock and conversion routes), which makes the biomass thermochemical process much less competitive and alluring. Another promising technical route of hydrogen production on a large scale is by the electrolysis of water [140]. However, converting higher-grade electrical energy into relatively lower-grade chemical energy, such as hydrogen and oxygen, is found to be contradictory to the practice of energy cascade utilization, let alone it being more mature and cost-effective to store and transport electricity compared to the hydrogen.

Apart from conventional centralized hydrogen production, on-site and decentralized small-scale hydrogen generation, which possesses the advantages of lowering the prices of transport and onsite utilization of non-usable biomass with high water content, has begun to attract more and more interest. The biological hydrogen generation process is found to be perfectly suitable for small-scale and decentralized hydrogen production using those non-usable biomasses with high water content, with the cost of hydrogen production varying from 10 to 20 U.S.$/GJ, which could be further improved via the R&D impetus in the foreseeable future. Holladay et al. recently made a comparison among different technologies for hydrogen generation synoptically [56] and the results are summarized in Table 6.

**Table 6.** The comparisons of different technical routes for hydrogen production, and their effectiveness.

| Route | Feedstock | Energy Efficiency/% |
|---|---|---|
| SR | Hydrocarbons | 70–85 [a] |
| POX | Hydrocarbons | 60–75 [a] |
| ATR | Hydrocarbons | 60–75 [a] |
| Plasma reforming | Hydrocarbons | 9–80 [b] |
| Pyrolysis | Coal | 50 [a] |
| Co-Pyrolysis | Coal + Waste material | 80 [a] |
| Photolysis | Solar + water | 0.5 [c] |
| DF | Biomass | 60–80 [d] |
| Photofermentation | Biomass + Solar | 0.1 [e] |
| Microbial electrolysis | Biomass + Electric | 78 [f] |
| PWS | Water + Solar | 12.4 [g] |

PWS photo-electrochemical water-splitting. [a] Thermal efficiency based on the higher heating values; [b] Does not include hydrogen purification; [c] Conversion of solar energy to hydrogen by water-splitting excluding hydrogen purification; [d] Theoretical maximum of 4 mol $H_2$ for 1 mol of glucose; [e] Conversion of solar energy to hydrogen by organic materials excluding hydrogen purification; [f] Total energy efficiency including applied voltage and energy in the substrate, excluding hydrogen purification; [g] Conversion of solar energy to hydrogen by water-splitting excluding hydrogen purification.

Clearly, BHP using biomass as feedstock shows very appealing effectiveness, let alone if further considering competitive and beneficial characteristics, such as the reduced environmental impact, and relative simplicity in operation compared with the thermochemical and electrochemical processes.

In addition, compared with photo-fermentation, DF presents very high efficiency, has a lower footprint, and is independent of solar energy. Therefore, it is envisioned that the effective approach of enhancing the activities of biocatalysts for BHP via DF is pivotal for highly efficient hydrogen production.

## 6. Future Perspectives

It is apparent that the enhancement of BHP during DF by process intensification and optimization has begun to approach its technical bottleneck at the current stage. From an energy cascade utilization and material recycling and reused perspective, the future for hydrogen production needs to implement multistage processes to further maximize the harvesting of solar energy [141]. The schematic diagram of a multistage procedure, comprised of four or five different steps or approaches, is proposed in Figure 5. In this system, the feeding flows of this multistage process are solar lights, renewable biomass, and water, and outflows are produced hydrogen gas, oxygen gas, and processed biomasses that could be further converted into value-added organic fertilizer [142]. Within this multistage conversion process, the hydrogen production is initiated by photo-fermentation and photocatalysis (solar water-splitting) with feeding-water and organic substrates. Within the system, the cascade utilization of organic substrates could further maximize the hydrogen production in each individual processing step. Theoretically, it is possible to acquire a maximum hydrogen production rate of 12 moles of hydrogen from 1 mole of substrate (glucose) through this combined approach, using purple non-sulfur photosynthetic bacteria and anaerobic bacteria by integrating DF with the photo-fermentations [143]. In addition, the proposed process also integrates the photocatalysis and micro-electrolysis processes for the sake of maximizing hydrogen productions of the entire process. The challenges of this proposed integrated process lies in: (a) the pH swing between the steps of the photofermentation stage, where ammonia will be generated continuously during the photofermentation catalyzed by nitrogenase, and the nearly neutral pH value of DF; (b) how to best optimize the feeding concentration of organic substrates (C/N/O ratio) [144] and control different metabolic pathways on the level of genetic expressions, as this will significantly affect the level of genetic diversity expressions during the fermentations [145–147]; (c) eco-friendly access to the water available.

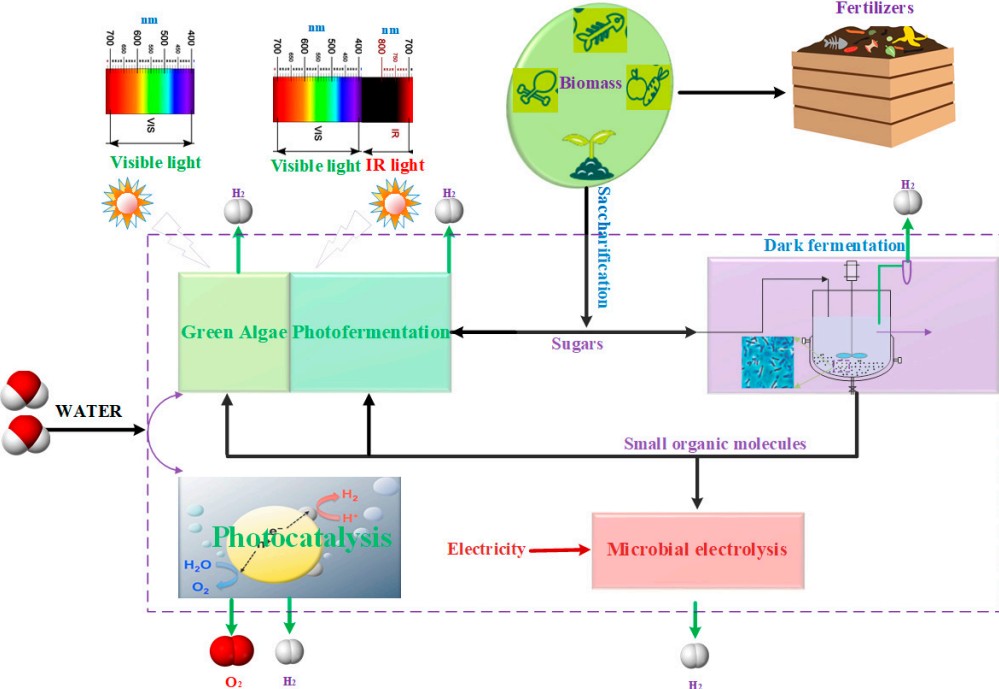

**Figure 5.** Simplified conceptual illustration diagram of an integrated and multi-coupling process of BHP (biohydrogen production) with photocatalysis using biomass as the substrate.

## 7. Conclusions

In this paper, biological hydrogen generation produced from renewable bio-resources was found to be a practical route for hydrogen production. Among different BHP routes, the DF has been found to be a practical approach in BHP, especially when it is enhanced by chemical addition. Among the different approaches of chemical addition to improve the activity of hydrogenase, the addition of NPs (Ni, Fe) was found to be relatively more efficient due to its direct effects of facilitating the electron transport between the ferredoxin and the hydrogenase. The order of effectiveness in enhancing the activities of hydrogenase on the basis of substrate conversion efficiency ($Y_{H2/S}$) and hydrogen evolution rate (HER) follows the order of metal NPs > metal monomers/metal ions/metal oxides > other metals (other than Ni, Fe). In order to make the BHP process more feasible and economical enough for industrial applications, future endeavors should focus on the optimized integration of different hydrogen production processes with the energy cascade utilization and material recycling and recovering. By appropriately integrating different approaches, it is potentially possible to approach the theoretical maximum hydrogen yield (12 mol $H_2$ per 1 mol Glucose consumption). These novel approaches of process intensifications, and integration and appropriate combination of several hydrogen generation processes, such as photocatalysis, photofermentation, and DF processes, will eventually facilitate large-scale curbing of the emission footprint and the cost of BHP in the foreseeable future.

**Funding:** The authors would like to appreciate the funding support from funding support from National Key R&D Program of China (2018YFC1903500), Edith Cowan University for staff support grant and staff excellent awards grant. The Faculty Inspiration Grant of University of Nottingham and Qianjiang Talent Scheme-Grant/Award Number: QJD1803014 are also highly appreciated.

**Acknowledgments:** The Acid-Based coupled production group at Institute of Process Engineering Chinese Academy of Science is also highly appreciated. The critical and insightful comments raised from three anonymous reviewers in significantly improving the quality of this work were also highly appreciated.

**Conflicts of Interest:** All authors have no conflict of interest in this work.

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
