# Peer review of "A Review of the Enhancement of Bio-Hydrogen Generation by Chemicals Addition"

_catalysts, doi:10.3390/catal9040353_

Reviewer 1 Report

This review from Yong Sun and colleagues present the state of the art of the addition of chemicals/metals in the biohydrogen production.

Although there is a number of similar works already published, this review finds its own space in the specific literature since it is mainly focused of dark fermentation.

The work is clear, well presented and, in my opinion, it could be of general interest.

The references presented are adequate for this kind of work and provide enough support regarding the role of chemicals addition in the production of biohydrogen.

The English used to draft the manuscript should be reviewed by a native speaker to adjust lexicon and expressions. There are parts of the manuscript in which the English used is good, and others with a lot of mistakes spread across the text, which of course needs a serious editing (see attached PDF file of the manuscript).

I would change the title in the following: “Enhancement of bio-hydrogen generation by affecting biocatalyst hydrogenase: role of chemicals addition.”

I recommend to publish this paper after minor revision.

Author Response

We appreciated reviewer’s comments, and those addressed comments have been responded in the manuscript accordingly with highlighting in blue (Line 3, Line 25, 26, 65, 221, 234, 503).

Reviewer 2 Report

Review catalyst-471093 for the Authors

Extremely interesting review of literature on receiving enhancement of bio-hydrogen (BHP) generation by affecting biocatalyst hydrogenase. The topic is extremely topical due to the search of exploring alternative renewable energy. The Authors have carried out with extreme care and meticulousness a review on different bio-hydrogen production  pathways, and the most important enzymes involved in this processes. However, the main focus of discussion of the Authors in this work was of the dark fermentation process  for the enhancement of BHP by chemical additions. Transparency of the review was added by literature data collected in six tables. In addition to the two figures transferred from the cited literature, the authors illustrated of this manuscript with their own diagrams and graphs (Figures 3-5). This procedure significantly helps the reader to understand the complexity of the topic discussed in this work. Certainly, this is an additional attribute of this work that deserves to be underlined.

In the reviewed work some elements need to be supplemented. Page 7 line 239 - should be Table 2 not Table 3.  Fd abbreviation concerning of ferredoxin used in Figure 3 and  Eqs (3-4) should be was given in the text (page 4, line 143).

Author Response

Extremely interesting review of literature on receiving enhancement of bio-hydrogen (BHP) generation by affecting biocatalyst hydrogenase. The topic is extremely topical due to the search of exploring alternative renewable energy. The Authors have carried out with extreme care and meticulousness a review on different bio-hydrogen production  pathways, and the most important enzymes involved in this processes. However, the main focus of discussion of the Authors in this work was of the dark fermentation process  for the enhancement of BHP by chemical additions. Transparency of the review was added by literature data collected in six tables. In addition to the two figures transferred from the cited literature, the authors illustrated of this manuscript with their own diagrams and graphs (Figures 3-5). This procedure significantly helps the reader to understand the complexity of the topic discussed in this work. Certainly, this is an additional attribute of this work that deserves to be underlined.

In the reviewed work some elements need to be supplemented. Page 7 line 239 - should be Table 2 not Table 3.  Fd abbreviation concerning of ferredoxin used in Figure 3 and  Eqs (3-4) should be was given in the text (page 4, line 143).

We appreciated reviewer’s comments, and those addressed comments have been responded in the manuscript accordingly with highlighting in blue.

Reviewer 3 Report

The review proposes to cover the methods to enhance molecular hydrogen production by focusing on few selected variables that affect hydrogen production via NiFe hydrogenases. 

NiFe are  surely important in the whole balance of hydrogen production but it should be outlined that as hydrogen producing catalysts they have low turnover frequency and the bias is often towards hydrogen consumption more than production. Also the role of nitrogenase should be discusses in terms of productivity. The amount of hydrogen produced is really low compared to other systems. 

There are some major remarks and flaws to be addressed:

As a general remark, although the focus might be interesting, at least some indication on the role of FeFe hydrogenases as the main hydrogen producing catalysts should be mentioned, as well as the role of microbial consortium in waste degrading process in DF. Also In several occasion in the text the authors themselves cite a process (i.e. Chlamydomonas or Clostridia based processes) which is largely or solely influenced by FeFe hydrogenases: this should be stated and discussed. Below are some key papers that should be mentioned for both aspects:

1.    Mechanisms for hydrogen production by different bacteria during mixed-acid and photo-fermentation and perspectives of hydrogen production biotechnology Critical Reviews in Biotechnology [0738-8551] Trchounian, Armen 2015 vol:35 fasc:1 pag:103 -113

2.    Changes in hydrogenase genetic diversity and proteomic patterns in mixed-culture dark fermentation of mono-, di- and tri-saccharides Quéméneur, M., Hamelin, J., Benomar, S., (...), Steyer, J.-P., Trably, E. 2011International Journal of Hydrogen Energy 36(18), pp. 11654-11665

3.    Arizzi, M., Morra, S., Pugliese, M., Gullino, M.L., Gilardi, G., Valetti, F. Biohydrogen and biomethane production sustained by untreated matrices and alternative application of compost waste (2016) Waste Management, 56, pp. 151-157. 

4.    Morra, S., Arizzi, M., Allegra, P., La Licata, B., Sagnelli, F., Zitella, P., Gilardi, G., Valetti, F. Expression of different types of [FeFe]-hydrogenase genes in bacteria isolated from a population of a bio-hydrogen pilot-scale plant (2014) International Journal of Hydrogen Energy, 39 (17), pp. 9018-9027.

5.    Investigation of hydrogenase molecular marker to optimize hydrogen production from organic wastes and effluents of agro-food industries Biotechnologie, Agronomie, Société et Environnement [1370-6233] Hamilton, C 2010 vol:14 fasc:SPEC. ISSUE 2 pag:574 -575

6.    Hamilton, C., Calusinska, M., Baptiste, S., Masset, J., Beckers, L., Thonart, P., Hiligsmann, S. Effect of the nitrogen source on the hydrogen production metabolism and hydrogenases of Clostridium butyricum CWBI1009 (2018) International Journal of Hydrogen Energy, 43 (11), pp. 5451-5462. 

7.    Masset, J., Calusinska, M., Hamilton, C., Hiligsmann, S., Joris, B., Wilmotte, A., Thonart, P. Fermentative hydrogen production from glucose and starch using pure strains and artificial co-cultures of Clostridium spp. (2012) Biotechnology for Biofuels, 5, art. no. 35, 

Also some assumptions on the role of NiFe hydrogenases are questionable.

At line 152 either the text is not clear and it induces to some misunderstanding or there is a wrong statement: “As the nickel is the main component in the biocatalyst hydrogenase metal cofactor center, the existence of nickel within the cell of microalgae will inevitably affect the hydrogen evolution during the biophotolysis”.  If we are speaking hydrogen production in microalgae, as stated, and as suggested by the species cited on line 150 (Chlamydomonas reinhardtii, Chlorella fusca, Scenedesmus obliquus, Chlorococcum littorale and Platymonas subcordiformis) please note that NiFe hydrogenases are not present in eukarya (please compare Peters et al., Biochimica et Biophysica Acta 1853 (2015) 1350–1369).  The authors should correct and clarify. The role of NiFe is instead relevant in cyanobacteria, where FeFe hydrogenases have not been reported yet, but cyanobacteria are not cited in the sentence above.

In Figure 3 the legend states: The conceptual illustration of the bio-hydrogen generation pathways a) biophotolysis, where PSI represents photosynthesis system 1, PSII represents photosynthesis system 2. b) photofermentation, c) DF. 134. Please make sure that the illustration is formally correct: for instance the water splitting occurs in PSII, while in the picture this is not represented. The over simplification of the picture leads to incorrect statements. This is also partially true for figure 5.

Minor remarks: please revise English (as examples)

Line 22 due its compelling (due toits compelling)

Line 65 …the nickel contained biocatalyst hydrogenase, (nickel-containingbiocatalyst hydrogenase)

Author Response

Reviewer 3

The review proposes to cover the methods to enhance molecular hydrogen production by focusing on few selected variables that affect hydrogen production via NiFe hydrogenases. 

NiFe are surely important in the whole balance of hydrogen production but it should be outlined that as hydrogen producing catalysts they have low turnover frequency and the bias is often towards hydrogen consumption more than production. Also the role of nitrogenase should be discusses in terms of productivity. The amount of hydrogen produced is really low compared to other systems. 

First of all, we do appreciate for reviewer’s insightful comments and provided references that will surely improve entire quality of this work. The misleading ‘nickel contained hydrogenase’ has been revised, in fact,  In this work, our ultimate focus is to review the chemicals additions in enhancing the biohydrogen generation during dark fermentation, which directly or indirectly affect the activity of hydrogenase (this could be either [NiFe] or [FeFe] depending on the different species that were cultivated). Therefore, the title of this paper has been revised accordingly.

In addition, some further revision to avoid confusion has been made throughout the manuscript:

Introduction part: line 25-26, line 70-74.

The role of low efficiency nitrogenase: line 103-108

What is more, the table 1 is also revised, which removed the confusing column of ‘nickel contained catalyst hydrogenase’. 

There are some major remarks and flaws to be addressed:

As a general remark, although the focus might be interesting, at least some indication on the role of FeFe hydrogenases as the main hydrogen producing catalysts should be mentioned, as well as the role of microbial consortium in waste degrading process in DF.

The role of [FeFe] hydrogenase has been added into the manuscript together with the waste degradation process during DF (the corresponding references [26], [142] and [145] were also cited). The detailed explanation of [FeFe] and [NiFe] were added into Line 115-122.

Also Figure 2 is revised with addition of a molecular structure of [FeFe] hydrogenase.

Also In several occasion in the text the authors themselves cite a process (i.e. Chlamydomonas or Clostridia based processes) which is largely or solely influenced by FeFe hydrogenases: this should be stated and discussed. Below are some key papers that should be mentioned for both aspects:

Indeed, we would like to focus on hydrogenase but not specific on either the [FeFe] type hydrogenase or [NiFe] type hydrogenase during the biophotolysis in the context. In addition, the suggested good references in this topics were added accordingly (reference [26] and [27]). In addition, the suggested references are surely helpful in tackling the confusions that arose from the first draft. They are indeed very useful and it will be beneficial to be included into this review paper so that future potential readers will be able to have much clearer ideas of the topic discussed in this work. All these following references have been added in different part of manuscript. 

1.    Mechanisms for hydrogen production by different bacteria during mixed-acid and photo-fermentation and perspectives of hydrogen production biotechnology Critical Reviews in Biotechnology [0738-8551] Trchounian, Armen 2015 vol:35 fasc:1 pag:103 -113

[141]

2.    Changes in hydrogenase genetic diversity and proteomic patterns in mixed-culture dark fermentation of mono-, di- and tri-saccharides Quéméneur, M., Hamelin, J., Benomar, S., (...), Steyer, J.-P., Trably, E. 2011International Journal of Hydrogen Energy 36(18), pp. 11654-11665

[145] 

3.    Arizzi, M., Morra, S., Pugliese, M., Gullino, M.L., Gilardi, G., Valetti, F. Biohydrogen and biomethane production sustained by untreated matrices and alternative application of compost waste (2016) Waste Management, 56, pp. 151-157. 

[142]

4.    Morra, S., Arizzi, M., Allegra, P., La Licata, B., Sagnelli, F., Zitella, P., Gilardi, G., Valetti, F. Expression of different types of [FeFe]-hydrogenase genes in bacteria isolated from a population of a bio-hydrogen pilot-scale plant (2014) International Journal of Hydrogen Energy, 39 (17), pp. 9018-9027.

[26]

5.    Investigation of hydrogenase molecular marker to optimize hydrogen production from organic wastes and effluents of agro-food industries Biotechnologie, Agronomie, Société et Environnement [1370-6233] Hamilton, C 2010 vol:14 fasc:SPEC. ISSUE 2 pag:574 -575

[144]

6.    Hamilton, C., Calusinska, M., Baptiste, S., Masset, J., Beckers, L., Thonart, P., Hiligsmann, S. Effect of the nitrogen source on the hydrogen production metabolism and hydrogenases of Clostridium butyricum CWBI1009 (2018) International Journal of Hydrogen Energy, 43 (11), pp. 5451-5462. 

[146]

7.    Masset, J., Calusinska, M., Hamilton, C., Hiligsmann, S., Joris, B., Wilmotte, A., Thonart, P. Fermentative hydrogen production from glucose and starch using pure strains and artificial co-cultures of Clostridium spp. (2012) Biotechnology for Biofuels, 5, art. no. 35, 

[147]

Also some assumptions on the role of NiFe hydrogenases are questionable.

At line 152 either the text is not clear and it induces to some misunderstanding or there is a wrong statement: “As the nickel is the main component in the biocatalyst hydrogenase metal cofactor center, the existence of nickel within the cell of microalgae will inevitably affect the hydrogen evolution during the biophotolysis”.  If we are speaking hydrogen production in microalgae, as stated, and as suggested by the species cited on line 150 (Chlamydomonas reinhardtii, Chlorella fusca, Scenedesmus obliquus, Chlorococcum littorale and Platymonas subcordiformis) please note that NiFe hydrogenases are not present in eukarya (please compare Peters et al., Biochimica et Biophysica Acta 1853 (2015) 1350–1369).  The authors should correct and clarify. The role of NiFe is instead relevant in cyanobacteria, where FeFe hydrogenases have not been reported yet, but cyanobacteria are not cited in the sentence above.

The ambiguity description “As the nickel is the main component in the biocatalyst hydrogenase metal cofactor center, the existence of nickel within the cell of microalgae will inevitably affect the hydrogen evolution during the biophotolysis” has been removed. In fact, as reviewer has pointed out that, it is incorrect to address that NiFe exist in eukarya. In this work, our focus is review the chemicals additions in enhancing the biohydrogen generation, which could potentially indirectly affect the activity of hydrogenase, which could be either [NiFe] or [FeFe] depending on the different species that were cultivated. The corresponding references has been added into the manuscript. In addition, Table 1 is also revised, which removed the confusing column of ‘nickel contained catalyst hydrogenase’. In addition, the suggested good reference (Biophysica Acta 1853 (2015) 1350–1369) was also cited in the revision in reference [27].   

In Figure 3 the legend states: The conceptual illustration of the bio-hydrogen generation pathways a) biophotolysis, where PSI represents photosynthesis system 1, PSII represents photosynthesis system 2. b) photofermentation, c) DF. 134. Please make sure that the illustration is formally correct: for instance the water splitting occurs in PSII, while in the picture this is not represented. The over simplification of the picture leads to incorrect statements. This is also partially true for figure 5.

The incorrect expression of Figure 3 is revised accordingly, of which the photo water splitting should occur in PSII. In Figure 5, it is a simplified schematic diagram for a multi-staged hydrogen production conceptual process.

Minor remarks: please revise English (as examples)

Line 22 due its compelling (due toits compelling)

Line 65 …the nickel contained biocatalyst hydrogenase, (nickel-containingbiocatalyst hydrogenase)

All manuscript has been carefully proof read to minimize the typos  

Round  2

Reviewer 3 Report

The authors have addressed all my concerns and I am happy for the work to be published as it is in the present form.